# Comparison of Asymmetry between Perceptual, Ocular, and Postural Vestibular Screening Tests

**DOI:** 10.3390/brainsci13020189

**Published:** 2023-01-23

**Authors:** Timothy R. Macaulay, Scott J. Wood, Austin Bollinger, Michael C. Schubert, Mark Shelhamer, Michael O. Bishop, Millard F. Reschke, Gilles Clément

**Affiliations:** 1KBR, Houston, TX 77058, USA; 2Neuroscience Laboratory, NASA Johnson Space Center, Houston, TX 77058, USA; 3Department of Otolaryngology-Head and Neck Surgery, Johns Hopkins University School of Medicine, Baltimore, MD 21205, USA

**Keywords:** vestibular asymmetry, head impulse test, Fukuda stepping test, unilateral centrifugation, subjective visual vertical, vertical and torsional alignment nulling

## Abstract

Background: A better understanding of how vestibular asymmetry manifests across tests is important due to its potential implications for balance dysfunction, motion sickness susceptibility, and adaptation to new environments. Objective: We report the results of multiple tests for vestibular asymmetry in 32 healthy participants. Methods: Asymmetry was measured using perceptual reports during unilateral centrifugation, oculomotor responses during visual alignment tasks, vestibulo-ocular reflex gain during head impulse tests, and body rotation during stepping tests. Results: A significant correlation was observed between asymmetries of subjective visual vertical and verbal report during unilateral centrifugation. Another significant correlation was observed between the asymmetries of ocular alignment, vestibulo-ocular reflex gain, and body rotation. Conclusions: These data suggest that there are underlying vestibular asymmetries in healthy individuals that are consistent across various vestibular challenges. In addition, these findings have value in guiding test selection during experimental design for assessing vestibular asymmetry in healthy adults.

## 1. Introduction

Bilateral asymmetry is normal across various human systems, including the inner ears. There are likely structural differences in the otolith organs of each person, leading to slightly different sensitivities in vestibular sensing across the midline. These small, underlying asymmetries between the left and right otolith organs can cause head tilt, ocular torsion, and skew deviation [1]; postural imbalance [2]; idiopathic scoliosis [3]; and motion sickness [4]. In normal, healthy individuals, this naturally occurring peripheral vestibular asymmetry is compensated for by central processes in a 1G environment; however, symptoms often manifest in situations that challenge the vestibular system.

Various screening tests for vestibular disorders can reveal asymmetry, such as asymmetry of gain and time constant of the vestibulo-ocular reflex [5], directional preponderance during unilateral centrifugation [6], bias in the subjective visual vertical [7], ocular misalignment [8], and directional deviation on the Fukuda stepping test [9,10]. Although these are not direct measures of vestibular function and symmetry, they do allow for comparisons in the direction and magnitude of asymmetry across tests.

Understanding this vestibular asymmetry is important because perceptual tilt, head and body tilt, vertical misalignment of the visual axes (skew deviation), and ocular torsion could be due to vestibular tone imbalance caused by a unilateral lesion to the graviceptive vestibular pathways [11]. Vestibular asymmetries can also contribute to balance dysfunction [12] and falls in the elderly [13], and some authors have suggested that vestibular asymmetry may be predictive of motion sickness [14,15,16]. A recent study using inner ear magnetic resonance imaging showed a larger morphological asymmetry of the bilateral vestibular organs in individuals with high motion sickness susceptibility [17]. Similarly, astronauts with higher otolith functional asymmetry are more prone to space motion sickness [18].

The purpose of this exploratory study was to determine the extent to which normal, healthy individuals exhibit asymmetries in several vestibular screening tests. We measured the asymmetry of subjective visual vertical during unilateral centrifugation of the right and left inner ear and the asymmetry of the vestibulo-ocular reflex during rotational (yaw) head impulses to the right and the left. We then compared these asymmetries with vertical and torsional ocular misalignments and the deviation of posture during a stepping test. Finally, we examined the correlations between these measurements. We hypothesized that consistent directional asymmetries are observed across multiple vestibular tests due to underlying asymmetries within the peripheral vestibular system of normal, healthy adults and are subject to a common central compensation.

## 2. Materials and Methods

### 2.1. Participants

Thirty-two healthy participants (18 females, 14 males; 24–61 years, mean ± SD; 38.6 ± 9.2 years) participated in this study located in the Neuroscience Laboratory at NASA Johnson Space Center. All procedures were approved by the NASA Institutional Review Board and were performed in accordance with the ethical standards of the 1964 Declaration of Helsinki. All participants confirmed that they had no known vestibular/oculomotor abnormalities or neuromuscular impairments and provided written informed consent before participating in the study. Twenty-six participants were right-handed, and twenty-two participants had a right visual dominance. There were no participant drop-outs or severe adverse events.

### 2.2. Perceptual Tilt

Participants were seated upright in a rotating chair (Neurolign Dx NOTC, Neurolign Technologies; Toronto, ON, Canada) with their heads oriented in the naturally erect position. Head position was maintained via adjustable foam pads positioned on the temples. A four-point harness restrained the shoulders and torso, and additional straps provided restraint to the legs and feet. Noise reduction headphones helped minimize extraneous orientation cues and enabled two-way audio communication between the participant and operator. The door to the rotating chair enclosure (6 ft diameter circular environment) was closed to ensure that testing was performed in complete darkness.

A stationary subjective visual vertical (SVV) assessment was performed with the chair immobile to familiarize participants with the test and confirm normal perception of verticality [19]. Participants were instructed to orient a laser light line (wavelength: 620–690 nm; output power: <1 mW class II laser) vertically relative to Earth using thumb buttons on top of the chair’s handles. Six trials were performed with the line starting at ±12°, ±15°, and ±20° in a random order.

During unilateral centrifugation, there were 5 SVV sets, each consisting of six trials with laser light lines that randomly started at ±12–22°. Participants were instructed to orient the lines vertically relative to Earth just as they did during the stationary SVV assessment. The first set of SVV lines (Center SVV) appeared 15 s after the chair reached a rotation velocity of 300 °/s in a clockwise direction (30 s ramp up). The chair then translated off-center to the right or left (randomly assigned) while maintaining constant rotation velocity. The translation distance was 38.5 mm, corresponding to a centripetal acceleration at the vestibular organ of 0.107 G. The second set of SVV lines appeared 6 s after the translation (Right SVV or Left SVV). The chair then translated back to center position, and the third set of SVV lines appeared 6 s after the translation. The fourth and fifth sets of SVV lines followed the same pattern as the second and third sets but with the chair translating in the opposite direction. After the fifth set of SVV lines, the chair slowed down to a stop.

Seventeen participants were translated left first then right. While rotating off-center, the gravito-inertial acceleration (GIA, i.e., the vector summation of gravity and centripetal acceleration) was tilted 6.09° relative to Earth vertical. Normal SVV responses were expected to be approximately equal in magnitude but opposite in direction to the GIA tilt with respect to vertical [20]. After adjusting the SVV lines, participants were asked to verbally estimate the direction and degree of perceived roll tilt.

For the calculation of the SVV and verbal report (VR) asymmetries, the measurements obtained when the chair was on-center (Center SVV, Center VR) were subtracted from the measurements obtained when the chair had translated to the right (Right SVV, Right VR) and left (Left SVV, Left VR). The SVV asymmetry index was then calculated as 100 × (Right SVV − Left SVV)/(Right SVV + Left SVV). The VR asymmetry index was then calculated as 100 × (Right VR − Left VR)/(Right VR + Left VR). Left and Right measures were of opposite polarity, so absolute values were used in the calculation. Positive SVV or VR asymmetry indexes indicate a greater response when the right vestibular organs were stimulated by the centripetal acceleration. Absolute values of these SVV and VR asymmetry indexes were also used for group statistics to represent the total magnitude of asymmetry.

### 2.3. Perceptual Vertical and Torsional Alignment

After the rotating chair stopped, participants remained seated but were allowed to loosen straps and restraints for comfort. A 12” active-matrix organic light-emitting diode (AMOLED) tablet computer (Galaxy TabPro S, Samsung; Suwon-si, Republic of Korea) was used to administer the vertical alignment nulling (VAN) and the torsional alignment nulling (TAN) tasks. The tablet was positioned 42 cm from the participant’s eyes using an arm fixture mounted to the chair.

A red line was designated as the stationary line and remained fixed on the screen. A blue line was designated as the moving line and could be repositioned by the participant either vertically (up and down) during VAN or in roll (clockwise and counterclockwise) during TAN. Participants wore colored lenses (BIAL Red-Blue 3D Glasses) so that they viewed the red line with only their left eye and the blue line with only their right eye. This removed any binocular cues for alignment [21]. Participants were instructed to adjust the moving line until it appeared to be aligned (vertically or torsionally) with the stationary line, then they were to tap a hidden box on the screen to indicate trial completion. Participants performed at least five practice trials of both VAN and TAN prior to data collection to ensure that participants understood the test instructions, that the room was dark (no light leaks), and that the participant could only see one colored line with each eye. Then, 22 trials of VAN were performed followed by 22 trials of TAN.

The amount by which the lines were separated from one another vertically (VAN) or rotated relative to one another (TAN) was used to calculate in degrees the perceptual measures of vertical and torsional binocular misalignment, respectively [8,21]. Positive VAN values indicated that the right line was positioned below the left line. Positive TAN values indicated that the right line was rotated clockwise with respect to left line. The absolute VAN and TAN values were also used for group statistics to represent the total magnitude of error.

### 2.4. Vestibulo-Ocular Reflex

Participants wore video-oculography binocular goggles (Neurolign DX Falcon, Neurolign Technologies; Toronto, ON) and were seated 213 cm from a target display. They were instructed to focus on a wall-fixed target that was straight ahead and to keep their neck muscles relaxed during a video head impulse test (vHIT) [22]. The operator rotated the participant’s head in rapid 20–30° impulses alternately to the right and left, pausing between each impulse, without returning to the center between impulses. The peak head velocity was 140–260 °/s. The vestibulo-ocular reflex (VOR) gain for each impulse (i.e., the average ratio of eye velocity overhead velocity from 40 °/s to 90% of peak head velocity) was calculated on multiple valid recordings for both leftward and rightward head impulses using the commercial analysis software (VEST 2019, Neurolign Technologies; Toronto, ON) [23]. Data from impulses with a VOR gain standard deviation ≥0.4 were excluded. The VOR gain asymmetry index was then calculated as (Right VOR gain − Left VOR gain)/(Right VOR gain + Left VOR gain). Positive VOR gain asymmetry index indicates a greater response during rightward head impulses than leftward. Absolute values were also used for group statistics to represent the total magnitude of asymmetry.

### 2.5. Fukuda Stepping Test

Participants started with their feet together standing on a labeled circular floor mat with their shod toes on a center dot. Then, they were instructed to march in place, bringing their feet up to normal stair stepping height, with their eyes closed and arms extended straight in front of them. Participants were asked to perform 50 steps (25 steps each leg) while counting their steps out loud and ending with their feet together [24]. An operator then placed a marker at the participant’s shod toes indicating both body position and the direction they were facing. The operator measured the participant’s displacement from the starting position (cm), deviation of body displacement (°), and body rotation (°) relative to their starting position. Leftward deviation and counterclockwise rotation were measured as positive. Absolute values were also used for group statistics to represent the total magnitude of deviation and rotation.

### 2.6. Statistical Analysis

Data from all 32 participants were reviewed and processed with R version 4.0.5 [25]. Abnormal data due to participant or operator error, or poor quality of eye movement recordings, were not included in the analysis (explained in the Section 3). Data are presented as means across participants ± standard deviations (SD), and the mean of the absolute values from each participant ± SD to represent the mean magnitude of error or asymmetry. After viewing the distribution of points for each of the six measures of interest (SVV Asymmetry, VR Asymmetry, VAN, TAN, VOR Gain Asymmetry, Body Rotation) and following up on potential outlier values, a non-parametric measure of rank correlation was selected as the metric for comparison. The Spearman’s rank correlation coefficient and accompanying *p*-values for the test of significance (alpha level of 0.05) were used to compare the asymmetry between the six measures of interest. In addition, simple paired *t*-test results were provided for comparing roll tilt when translated left versus when they were translated right. Due to the exploratory nature of the study, no adjustments were made for multiple comparisons, and all conducted tests of significance have been reported. Correlation values between the six measures were also used to create a network plot, which was utilized to help visually represent the relationships between the six measures of interest.

## 3. Results

### 3.1. SVV and VR Asymmetry

All participants completed the unilateral centrifugation test. However, SVV data from one participant were excluded due to a misunderstanding of the test instructions. VR data from three participants were not collected, and VR data from one participant were excluded due to a misunderstanding of the test instructions. On average, participants perceived greater roll tilt when translated left (SVV: 6.5° ± 4.8°, VR: 13.8 ± 12.0°) than when translated right (SVV: 4.7 ± 2.9°, VR: 12.0 ± 12.6°) (SVV: *p* = 0.006, VR: *p* = 0.245); paired *t*-test on the absolute values). The SVV asymmetry index was −12.8 ± 29.5% (absolute: 24.4 ± 20.1%). The VR asymmetry index was −9.6 ± 39.9% (absolute: 28.7 ± 25.1%).

The errors in perceived tilt relative to the GIA for each individual SVV trial and SR reports are presented in Figure 1A,B, respectively. On average for SVV, there was an overestimation of tilt when centrifugal acceleration was applied on the left ear and an underestimation of tilt when centrifugal acceleration was applied on the right ear. However, verbal reports indicated an overestimation of tilt for both conditions.

### 3.2. Ocular Alignment

All participants, except one, completed all 22 trials of VAN and TAN. One VAN trial from one participant was excluded due to a premature unintentional click to proceed to the next trial. The mean ocular misalignments were 0.017 ± 0.092° for VAN and 0.258 ± 0.666° for TAN (Figure 2). The absolute mean error for vertical and torsional ocular misalignments was 0.064 ± 0.069° and 0.512 ± 0.497°, respectively.

### 3.3. VOR Gain Asymmetry

All participants completed the vHIT. However, data from one participant were excluded due to poor data quality. VOR gain calculated for all the trials in 31 participants are shown in Figure 3. On average, VOR gain during head impulses to the right was 1.15 ± 0.22. VOR gain during head impulse to the left was 1.12 ± 0.17. The calculated VOR gain asymmetry index was +1.11 ± 7.04%, and the absolute VOR gain asymmetry index was 5.42 ± 4.45%.

### 3.4. Body Rotation

All participants completed the Fukuda stepping test. However, body rotation data from four participants were not collected. On average, during the completion of 50 steps, participants tended to move forward (77.0 ± 31.5 cm) and deviate to the right (−12.3 ± 27.5°) with a clockwise body rotation (−15.3 ± 53.0°) (Figure 4). The absolute mean displacement, deviation of displacement, and body rotation were 77.0 ± 31.5 cm, 23.6 ± 18.7°, and 38.3 ± 39.9°, respectively. Note that for all but one participant, the deviation of displacement and body rotation were in corresponding directions (e.g., a leftward deviation of displacement was accompanied by a counterclockwise body rotation).

### 3.5. Correlations

Table 1 presents a correlation matrix between the asymmetries of the responses to the various screening tests. There is a significant correlation between SVV and VR asymmetries. VAN is significantly correlated with TAN, VOR gain asymmetry, and body rotation. Finally, the VOR gain asymmetry is significantly correlated with body rotation. These relationships between measures are depicted visually in a network plot in Figure 5. The network connections (lengths and intensities) are based on their correlation strengths.

## 4. Discussion

### 4.1. Vestibular Screening Tests

Vestibular screening tests were used to challenge the vestibular systems of normal, healthy participants and reveal naturally occurring vestibular asymmetries that are normally compensated for by central processes. As we describe here, the mean values are within established norms, despite the limitations of having a small sample size and high variances. Data from all trials are presented in the figures above for visualizing these variances. The magnitudes of asymmetry are relatively small compared to patient populations, to which comparisons are made for context of how these tests are typically used in clinical settings. Although these are not direct measures of vestibular function/symmetry, some consistent directional asymmetries between participants suggest that there may be natural biases in the population [4], similar to left/right handedness.

Unilateral centrifugation of the vestibular organs at 0.107 G generated a GIA tilt of 6.09° relative to Earth vertical. The SVV indicated by the participants was close to the GIA tilt, but the verbal reports were about twice the tilt of the GIA. These results are in agreement with the literature [26,27]. The differences between the SVV and the verbal reports could be explained by the fact that the centripetal acceleration also induces a counterrotation of the eyes, which influences the SVV measurements [28]. SVV and VR have been shown to be altered during unilateral centrifugation in patients with unilateral vestibular deafferentation [6,26], benign paroxysmal positional vertigo [29], and undiagnosed dizziness [27], as well as in astronauts during spaceflight weightlessness [30]. Furthermore, SVV results are often correlated with clinical dizziness symptoms [31]. The asymmetry indexes suggest that participants perceived greater roll tilt when translated to the left, with a mean left/right asymmetry magnitude of about 25%.

Ocular alignment errors in the present study are also in agreement with previous studies using the same equipment and methods [8,21]. Ocular alignment errors can occur due to a lack of coordination between the oculomotor and the vestibular system. VAN and TAN are behavioral measures of oculomotor misalignment, which can arise in various ways. Damage to the utriculo–ocular pathway can cause an ocular tilt reaction (accompanied by head tilt) [1] and otolith asymmetry can be one cause of skew deviation [32], both shown to manifest in torsional vertical or ocular misalignment. Because the mean vertical and torsional ocular misalignments were both positive, it suggests that participants perceived, on average, the right line to be higher and more rotated counterclockwise than the blue line. The mean magnitudes of misalignment were less than 0.1° for vertical and about 0.5° for torsional results.

VOR gain and asymmetry were within the norms measured with the vHIT [33]. In fact, the mean left/right asymmetry magnitude was very low at about 5.4%. This was expected given the healthy participant population. Although high variability was observed with the presentation of data from all trials, this potential limitation is expected with field measures of VOR gain using automated pre-processing techniques. Patients with bilateral vestibular loss or acute unilateral loss typically present deficits in VOR gain, representing a loss of semicircular canal function [34]. VOR gain asymmetry is often observed in unilateral loss patients, such as after vestibular neuritis or unilateral vestibular deafferentation [35]; however, it is not always consistent with the symptoms observed or with caloric nystagmus asymmetry [36]. McGarvie et al. [33] suggest that asymmetries can be a systematic effect of geometry that occurs with single eye measurements, but the current study used combined left/right eye data to avoid this confounder.

The Fukuda stepping test results suggest that participants tend to rotate to the right and have a mean magnitude of body rotation around 38°. This test is designed to assess descending vestibulo-spinal control by measuring changes in body position that occur while stepping in place with the eyes closed. The degree of deviation and body rotation are more reliable indicators of vestibular loss than displacement [9]. Larger deviations and body rotations are seen in patients with unilateral vestibular lesions [37] and other peripheral vestibular diseases [38]. However, this test may be unreliable for screening peripheral vestibular asymmetry in patients with chronic disorders [39].

### 4.2. Correlations between Perceptual, Ocular, and Postural Tests

Strong correlations and close proximities in the network plot between SVV and VR, and between ocular misalignment, body rotation, and VOR gain, indicate that the various tests measure the same underlying asymmetry within our healthy participants. By contrast, weak correlations indicate that the perceptual tests might be measuring different aspects of vestibular asymmetry than the ocular/postural tests. A dissociation between perceptual and ocular responses is commonly observed during visual [40] or vestibular [41,42] stimulations. The rationale for such dissociation is that perception requires the integration of sensory signals over time, whereas oculomotor and postural responses are more reflex-like responses [43].

The correlation between the asymmetry of the subjective reports is not surprising given that they are both measurements of the perceived tilt relative to the subjective vertical, with or without vision. Similarly, VAN and TAN are both measurements of ocular misalignments and so are well correlated with each other. The correlation between the asymmetry of body rotation and VOR gain was also expected because these responses were in the same plane (yaw). The correlation between VAN and body rotation is not as obvious but may be due to common vestibular pathways. The VAN response includes otolith input [44] and a majority of otolith afference is used for vestibulospinal control [45,46]. Thus, the correlation between VAN and body rotation most likely relates to the shared convergence and innervation patterns of the canal and otolithic neurons onto the vestibulospinal system.

We were expecting the TAN response to be better correlated with SVV and VR because there are responses in the same plane (roll tilt). They also reflect the integration of multiple cues (physiological and perceptual). VAN/TAN, for example, is a functional assessment of misalignment: it is influenced by both motor fusion (extraocular muscle activity to align the eyes) and sensory fusion (cortical activity to perceptually merge retinal images even if motor fusion has not aligned them perfectly). Thus, their weak correlation is indeed surprising. An obvious difference between these tests, which might explain this discrepancy, is that VAN/TAN was performed only with the head upright and stationary, while SVV and VR involve centrifugation. Although VAN/TAN is performed in darkness so that latent misalignments due to otolith asymmetry might become manifested, it is likely that the central compensatory mechanisms that maintain alignment under normal conditions (1 g, head upright) are not sufficiently challenged in this test protocol [44]. This argues for VAN/TAN testing with head tilt.

## 5. Conclusions

These data suggest that there are underlying vestibular asymmetries within healthy adults that are consistent across various vestibular challenges. Although interpretations of our findings are constrained to normal, healthy populations, we hypothesize that vestibular asymmetries may be exacerbated by disease, aging, or extreme environments. With greater underlying asymmetries, the correlations between tests may also be larger. Thus, further research is necessary to understand how vestibular asymmetries manifest across these tests in other populations, such as patients with vestibular disorders or astronauts after g-transitions. This understanding can help us develop appropriate rehabilitation strategies for astronauts based on therapies currently used in patients with vestibular disorders.

These findings may also help in down-selecting standard vestibular tests for research protocols involving normal, healthy participants when considering time and equipment availability. Tests that are highly correlated, with asymmetry showing up in both tests, may provide redundant information. Given the high variance observed in the current study, future larger studies conducted to confirm the findings presented here would benefit from including test–retest reliability to determine the effects of within-subject variability. Performance of multiple tests (even if correlated with each other) can yield useful information, increase confidence, and provide clinical nuance. Nevertheless, there are circumstances where testing time, expertise, and apparatuses are limited. This includes research laboratories, astronauts in a spacecraft or in a habitat, and various field settings (sports, military operations, etc.).

## Figures and Tables

**Figure 1 brainsci-13-00189-f001:**
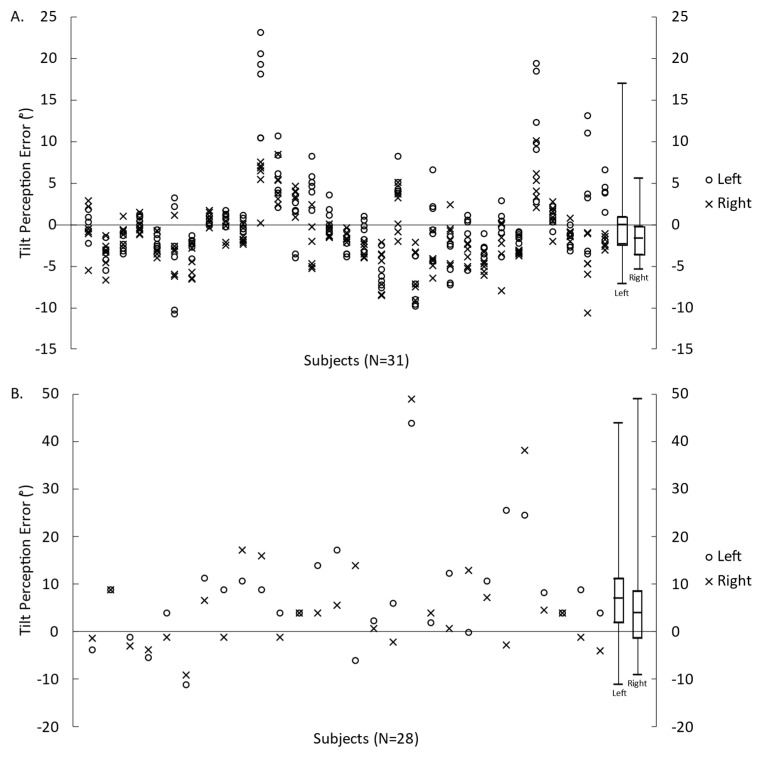
Error in tilt perception relative to the GIA measured by the SVV (**A**) and VR (**B**) for translations to the left and to the right. Symbols represent the measures for each participant, and box plots represent the minimum, first quartile, median, third quartile, and maximum values (calculated from each participant’s mean value).

**Figure 2 brainsci-13-00189-f002:**
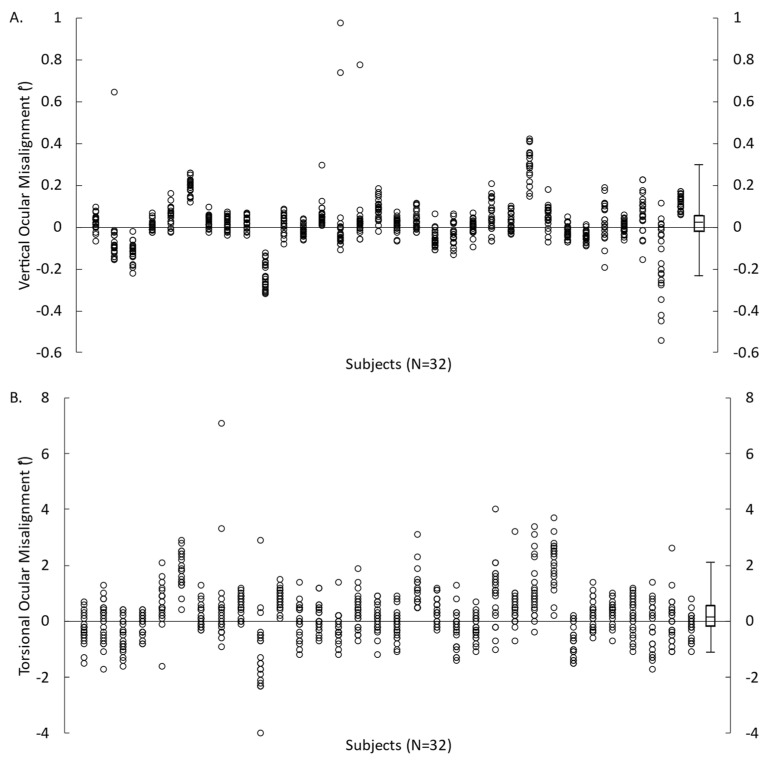
Ocular misalignment obtained from each VAN (**A**) and TAN (**B**) trial for 32 participants. Symbols represent the measures for each participant and box plots represent the minimum, first quartile, median, third quartile, and maximum values (calculated from each participant’s mean value).

**Figure 3 brainsci-13-00189-f003:**
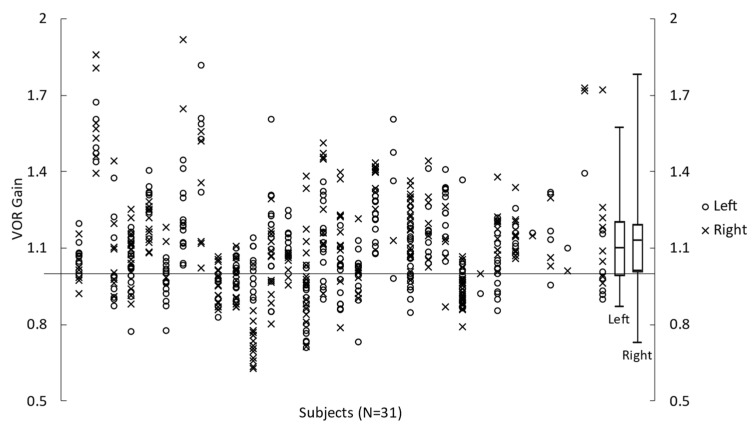
Left and Right VOR gain during vHIT impulses obtained for 31 participants. Symbols represent the measures for each participant, and box plots represent the minimum, first quartile, median, third quartile, and maximum values (calculated from each participant’s mean value).

**Figure 4 brainsci-13-00189-f004:**
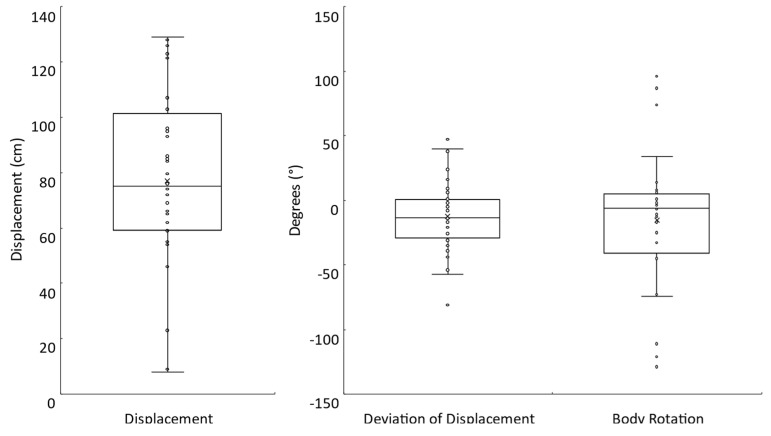
Fukuda stepping test. Box plots with individual participant data shown for the displacement (N = 32), deviation of displacement (N = 32), and body rotation (N = 28) measures.

**Figure 5 brainsci-13-00189-f005:**
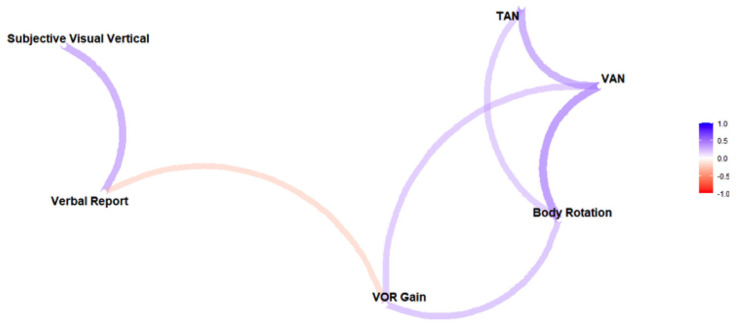
Network plot based on Spearman’s rank correlation coefficients, which suggests 2 possible groups of vestibular asymmetry measures: perceptual reports (on the left) and ocular–postural responses (on the right). Stronger correlations are represented by shorter, more intense (darker) paths. Colors indicate the sign (blue for positive and red for negative).

**Table 1 brainsci-13-00189-t001:** Correlation matrix between asymmetry of vestibular responses. Spearman’s rank correlation coefficient (*p*-value, without adjustment for multiple testing). * *p* < 0.05.

	SVV Asymmetry	VR Asymmetry	VAN	TAN	VOR Gain Asymmetry	Body Rotation
SVV asymmetry	1	0.473 (0.013) *	0.175 (0.346)	0.103 (0.582)	−0.154 (0.414)	0.017 (0.931)
VR asymmetry		1	0.017 (0.932)	0.203 (0.299)	−0.326 (0.097)	−0.276 (0.181)
VAN			1	0.480 (0.005) *	0.367 (0.043) *	0.541 (0.003) *
TAN				1	−0.152 (0.414)	0.364 (0.057)
VOR gain asymmetry					1	0.391 (0.043) *
Body rotation						1

## Data Availability

The data that support the findings of this study are available upon reasonable requests.

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
