# Peer review of "Comparison of Asymmetry between Perceptual, Ocular, and Postural Vestibular Screening Tests"

_brainsci, 2023, doi:10.3390/brainsci13020189_

Round 1

Reviewer 1 Report

Line 200: Please reword "unintentional end-of-trial indication" to clarify what you're trying to say here. 

Beautiful representation of the data in Figure 5. 

Lines 320-321: Be very cautious to jump your conclusions to the clinic populations of patients with vestibular disorders. The aim of this study was the detection of bias in those without vestibular disorders, for the important reasons you list. However, detecting bias in those with vestibular disorders requires a whole bunch of data not assessed here. You note in lines 285-286 that this is a limitation of this study, but I disagree if you stay with the population at hand, and try not to extend findings to a population you did not test. 

Reviewer 2 Report

Macaulay and colleagues present a well-written manuscript testing, and comparing tests of, vestibular imbalance in neurotypical, young, healthy adults. This work can serve to facilitate comparison between testing approaches, which may yield differing results and motivate divergent conclusions from the same sample population. While correlations between VOR asymmetry and Fukuda Stepping task results are not overly surprising, it is notable that some perceptual-motor tasks such as vertical and torsional alignment nulling correlate with these highly-automatic vestibular reflex tests, while others, such as subjective visual vertical, do not. These data have value in guiding test-selection during experimental design and/or clinical testing for vestibular function or asymmetry. 

I have grouped comments into major and minor concerns to help improve the manuscript. On the whole, I feel all points I raise here are relatively minor concerns. 

MAJOR CONCERNS: 

SAMPLE SIZE AS A LIMITATION. While I feel the sample size given is sufficient to draw preliminary conclusions about relationships between test performance, I would like the authors to note the relatively small sample when discussing limitations to this study. There appears to be a lot of variance in the data (magnitude of SD > mean for some measures), and the effects appear to be relatively small (<30% of variance between correlated measures). Given that all tests were within-participant examinations of vestibular function, it seems plausible that some results may differ upon replication. A note in the limitations acknowledging this possibility, and guiding future replication/interpretation, would be appropriate. 

FUKUDA TEST MEASURES. How were displacement and rotation measures obtained following the Fukuda Stepping test? Was specialized equipment used (e.g., motion capture or accelerometry)? Were anatomical landmarks consistently used? More information about the method would be appropriate. 

DATA INTERGRETATION FRAMEWORK. Both absolute and directional values were used for most variables but the authors do not give guidance on how different values should be interpreted. They are also not distinctly interpreted in the discussion. Some more detail on what different interpretations are drawn from these data (and/or need for both sets of measures), particularly in the discussion, would be appreciated. 

MONOR CONCERNS: 

Suggest you change ‘Subject’ to ‘Participant’. 

VAN/TAN practice. Said participants did ‘at least’ 5 trials. Does that mean some did more? Do they have data on range/average? 

NETWORK PLOT (lines 175-177) – not clear how this plot was constructed, and whether any specialized approaches were used (i.e., if done in R, which packages were used?). 

Please note parenthetically that data exclusions will be explained in the Results (Methods – Statistical Analyses; lines 164-166).

FIGURE 3 – Why show VOR gain and not VOR asymmetry? I would think asymmetry would be more relevant to the primary question of this paper.

DATA EXCLUSIONS - Body rotation (lines 219-220). Why were 5 participants excluded if the participants performed the task? If due to equipment malfunction, etc., please explain in the results (consistent with explanations given for other exclusions).

Reviewer 3 Report

This manuscript presents a range of vestibular assessments in a relatively small cohort of healthy individuals in order to assess the magnitude of vestibular asymmetry across these tests.

I have a number of reservations about the methodology, results, and conclusions drawn from the data:

1. The overall concept of this paper seems overly simplistic in its approach. The authors assume that the tests employed here allow for direct measures of vestibular function and symmetry, whereas in fact there is likely to be substantial asymmetry generated by the tests themselves owing to anatomical (not functional) differences across individuals. Such artefactual asymmetry cannot be comprehensively evaluated in a small group of healthy individuals, as evidenced by the fairly large standard deviations within each of the various tests.

2. Did the authors conduct any power calculations before conducting the study? It seems the study is substantially under-powered given the number of tests performed, and comparisons made. 

3. Relatedly, the authors state that: "no adjustments were made for multiple comparisons" due tot he exploratory nature of the study. There was no a priori mention that this was an exploratory study in the introduction, and in any case, the absence of adjustment for multiple comparisons will naturally negate the relevance of some of these statistically significant results.

4. Following from point 1, there are likely to be inherent biases related to the vestibular tests themselves. This is relevant primarily I believe for the Fukuda stepping tests, where a lot more than just vestibular function is being probed, given the complex neural circuitry involved in stepping, and the vHIT, where the test appears to have been conducted with standard binocular oculography...

5. Related to this latter point, the gain values presented for the vHIT are alarmingly large (Figure 3) for some individuals (and mostly to the left) - rather than reflecting vestibular function asymmetry I would argue that this may relate to the goggles (slippage?) and handedness of the examiner? Gains approaching 2 are always artefactual.

6. Whilst Figure 5 is pretty, I am not convinced this is the correct way to express the data, as these variables are not necessarily inter-related (again noting no corrections were made for multiple comparisons). 

7. How do the authors account for the correlation between VAN (roll) and body rotation (yaw)?

8. In the discussion, much of the early paragraphs deal with disease states, but this study was exploring asymmetry in the healthy population, so I think the discussion lacks focus. On page 9, lines 271 onwards, the authors discuss how high frequency vestibular responses may recover at different rates than lower frequency responses, but this ignores a large body of evidence that shows how vestibular function (regardless of frequency) does not correlate with symptom load (see Bronstein & Dieterich, Current Opinions Neurol 2019 for a review).

9. In the discussion (section 4.2), the authors mention the use of healthy individuals as a limitation of the study. Was this not the primary aim, to assess asymmetry in a healthy population?

10. I do not agree with the concluding remarks as I cannot see how one would practically utilise the data presented in this report. The argument that asymmetries in vestibular tests results equate directly to an asymmetry in vestibular function itself is not sufficiently robust.

Minor points:

The paper starts with "bilateral asymmetry" - what would 'unilateral asymmetry' be?

A reference is needed for the use of their oculography device for VOR testing (currently reference 22, but this is a reference dealing with vHIT more generally. 

Round 2

Reviewer 3 Report

the authors have addressed all comments, thank you